evolution, genetics, computational biology

genetic diversity, botanic gardens, conservation planning, seed banks, *ex situ* conservation, trees

**Author for correspondence:**
Sean Hoban
e-mail: shoban@mortonarb.org

# Taxonomic similarity does not predict necessary sample size for *ex situ* conservation: a comparison among five genera

Sean Hoban[1], Taylor Callicrate[2], John Clark[3], Susan Deans[4], Michael Dosmann[5], Jeremie Fant[6], Oliver Gailing[7,8], Kayri Havens[6], Andrew L. Hipp[1], Priyanka Kadav[8], Andrea T. Kramer[6], Matthew Lobdell[9], Tracy Magellan[10], Alan W. Meerow[10], Abby Meyer[11], Margaret Pooler[12], Vanessa Sanchez[10], Emma Spence[1], Patrick Thompson[13], Raakel Toppila[14], Seana Walsh[15], Murphy Westwood[9], Jordan Wood[6] and M. Patrick Griffith[10]

[1]Center for Tree Science, The Morton Arboretum, Lisle, IL, USA
[2]Species Conservation Toolkit Initiative, Chicago Zoological Society, Brookfield, IL, USA
[3]San Diego Zoo, San Diego, CA, USA
[4]Plant Biology and Conservation Program, Northwestern University, Evanston, IL, USA
[5]The Arnold Arboretum of Harvard University, Boston, MA, USA
[6]Negaunee Institute for Plant Conservation Science and Action, Chicago Botanic Garden, Glencoe, IL, USA
[7]Department of Forest Genetics and Forest Tree Breeding, University of Göttingen, Göttingen, Germany
[8]Michigan Technological University, Houghton, MI, USA
[9]The Morton Arboretum, Lisle, IL, USA
[10]Montgomery Botanical Center, Coral Gables, FL, USA
[11]Botanic Gardens Conservation International, US, San Marino, CA, USA
[12]USDA-ARS, US National Arboretum, Beltsville, MD, USA
[13]Auburn University, Auburn, AL, USA
[14]Squamish, Canada
[15]Department of Science and Conservation, National Tropical Botanical Garden, Kalāheo, HI, USA

(iD) SH, 0000-0002-0348-8449; JF, 0000-0001-9276-1111; SW, 0000-0002-4829-4488

Effectively conserving biodiversity with limited resources requires scientifically informed and efficient strategies. Guidance is particularly needed on how many living plants are necessary to conserve a threshold level of genetic diversity in *ex situ* collections. We investigated this question for 11 taxa across five genera. In this first study analysing and optimizing *ex situ* genetic diversity across multiple genera, we found that the percentage of extant genetic diversity currently conserved varies among taxa from 40% to 95%. Most taxa are well below genetic conservation targets. Resampling datasets showed that ideal collection sizes vary widely even within a genus: one taxon typically required at least 50% more individuals than another (though *Quercus* was an exception). Still, across taxa, the minimum collection size to achieve genetic conservation goals is within one order of magnitude. Current collections are also suboptimal: they could remain the same size yet capture twice the genetic diversity with an improved sampling design. We term this deficiency the 'genetic conservation gap'. Lastly, we show that minimum collection sizes are influenced by collection priorities regarding the genetic diversity target. In summary, current collections are insufficient (not reaching targets) and suboptimal (not efficiently designed), and we show how improvements can be made.

## 1. Introduction

*Ex situ* collections of plants and animals held in botanic gardens, arboreta, zoos and seed banks inspire and educate the public and provide material for scientific

**Table 1.** Taxa examined, distribution, reproductive biology and sampling.

| species | pollination | monoecious (M) or dieceous (D) | seed dispersal | pop size in situ[a] | pop size ex situ |
|---|---|---|---|---|---|
| *Hibiscus waimeae* A. Heller subsp. hannerae D. M. Bates | insect | M | gravity/water/unknown | 150–200 | 500–600 |
| *Hibiscus waimeae* A. Heller subsp. waimeae | insect | M | gravity/water/unknown | estimated 1000 | 100–200 |
| *Magnolia ashei* Weatherby | insect (beetles) | M | bird | approx. 3000 | approx. 50 |
| *Magnolia pyramidata* W. Bartram | insect (beetles) | M | bird | unknown | <50 |
| *Pseudophoenix ekmanii* Burret | insect (generalist) | M | gravity + animal | approx. 2000 | 91 |
| *Pseudophoenix sargentii* H. Wendland ex Sargent | insect (generalist) | M | gravity + animal | >2000 | 96 |
| *Quercus boyntonii* Beadle | wind | M | gravity + animal | 500–1000 | approx. 300 |
| *Quercus georgiana* M.A Curtis | wind | M | gravity + animal | >200 | approx. 200 |
| *Quercus oglethorpensis* Duncan | wind | M | gravity + animal | 1000 | <400 |
| *Zamia decumbens* Calonje, Meerman, M. P. Griff. & Hoese | unique obligate insect | D | gravity/unknown | <1000 | 205 |
| *Zamia lucayana* Britton | unique obligate insect | D | gravity/unknown | approx. 1000 | 244 |

[a]Pop size refers to the estimated or actual number of plants *in situ* and *ex situ*. Table 2 gives sample sizes that were genotyped for this study.

study and ecological restoration. These collections also help safeguard species from extinction, especially when species' native habitats become uninhabitable. Currently, 14 300 plant species are classified by the International Union for Conservation of Nature Red List as threatened with extinction (greater than 10% chance of extinction in less than 100 years) including 63% of cycads and 40% of conifers, but most plant species have not been assessed. Meanwhile, many millions of populations of mammal species have already disappeared owing to habitat loss [1], suggesting that a precipitous loss of genetic and trait diversity even before extinction occurs [2,3]. A scientifically informed strategy that integrates *in situ* and *ex situ* conservation approaches is recommended to prevent extinction and ensure that genetic diversity necessary for long-term species survival is maintained [4,5].

Governments and conservation organizations have committed to help safeguard genetic biodiversity. For example, all signatories of the Convention on Biological Diversity (https://www.cbd.int/sp/targets/) have committed to 'preventing genetic erosion' and 'safeguarding genetic diversity'. Meanwhile, the United Nations Sustainable Development Goal 2.5 (https://sustainabledevelopment.un.org/) focuses on agricultural biodiversity and food security via seed banks, and the Global Strategy for Plant Conservation (GSPC; https://www.cbd.int/gspc/targets.shtml) commits to preserving 70% of genetic diversity of crops, wild relatives and other economically or culturally valued species by the year 2020. Actions to date are far from sufficient [6,7], and perhaps as little as 3% of plant species have their genetic diversity sufficiently preserved *in situ* and 40% *ex situ* (assuming geographical range loss correlates to genetic diversity, [8]). Progress is being made *ex situ*, with 3200 botanic gardens worldwide preserving at least 105 000 species [9], and with many seedbanks conserving agrobiodiversity, especially domesticated species.

Unfortunately, excepting the most valuable crops, many *ex situ* collections contain only a few specimens per species, usually of limited geographical provenance [10,11]. Thus, many thousands of plant species probably lack sufficient genetic conservation to ensure species' long-term survival. (Note: here, we will use 'collection' as a noun to mean living specimens in a long-term archive (e.g. seed bank, botanic garden and arboreta); we use 'sampling' or 'acquisition' to refer to the action of collecting material from the wild (e.g. seeds and cuttings).)

There is a particular challenge for *ex situ* collections of species that are very difficult to conserve *ex situ* [12]. While seed banks are an efficient genetic safeguard for many plant species, about 20% of plant species are 'exceptional species' (*sensu* [13])—plants with recalcitrant seeds (seeds that cannot survive in standard seedbank conditions) or other sampling or storage challenges. Although cryopreservation is an option, it is very expensive, so they most frequently are maintained as living plants in collections. The challenge is capturing high genetic diversity in as few individuals as possible. A botanic garden might have resources to maintain a few to a few hundred individuals of some priority species, but not the thousands that seed banks can preserve (though collections of rare plants are often spread among numerous locations, i.e. the metacollection). Providing scientifically grounded recommendations for how many individuals to conserve, and how to acquire and manage them, would be a key scientific and practical contribution to ensure efficient (using resources wisely) and effective (meeting biodiversity targets) conservation.

Standard collection sizes have been established using simple probability equations and are frequently applied to many species (specifically the recommendation of [14] for sampling seed from 50 individuals to create an *ex situ* collection; see also [15]; table 1). However, recent research suggests that standard collecting guidelines might not be optimal practice for capturing

the maximum diversity across all species. Specifically, it is known that the amount of genetic diversity in a given set of individuals is influenced by the target species' biological traits (e.g. pollination mode, vegetative growth, frequency of flowering, seed dispersal and breeding system), as well as other factors such as phylogenetic history and demography (e.g. population fragmentation, historic and current ranges, and glacial refugia), which vary tremendously among species [16,17]. These factors influence how genetic diversity is allocated among seeds, individuals and populations across space, and thus, the effectiveness and efficiency of a given seed sampling strategy applied to different species. For instance, using models of key biological traits, Hoban & Strand [15] showed that species which exhibit overlapping age structure, short distance dispersal and high self-pollination rates will have less genetic diversity captured in a given collection size than species without those traits, such that more than five times as many individuals may be needed for some species than for others. Griffith *et al.* [18] investigated two cycad species (*Zamia*) and showed that less genetic diversity will be captured in the species with a lower frequency of flowering and fewer reproductive individuals. In spite of this foundational work, the question remains open as to what primary factors (such as shared traits within a genus) could lead to quantitatively similar collecting strategies.

Given the numerous factors affecting the ideal collection size across the diverse plant kingdom, an important advance would be to determine whether it is possible to generalize guidance for taxa in the same genus compared to other genera. *Will similar collection sizes for multiple species within a genus capture similar proportions of genetic diversity owing to shared phylogenetic history and characteristics such as overall size, growth habit, pollination biology, and seed dispersal?* A multi-taxa empirical study across the plant kingdom is needed to establish this knowledge but has never, to our knowledge, been performed. This knowledge is consequential for conservation decisions; if species within a genus can be conserved relatively similarly, then a few genetic studies within a genus could provide broadly applicable guidelines for all species in that genus, resulting in reliable advice for conservation without having to perform a genetic study in every target species.

In this paper, we present such a comparative study. We use genetic datasets and resampling algorithms to determine appropriate *ex situ* collection size in 11 woody, perennial taxa (nine species and two subspecies) from five genera spanning much of the seed plant tree of life (*Hibiscus*, *Magnolia*, *Pseudophoenix*, *Quercus* and *Zamia*). All are relatively long-lived, threatened species whose seeds cannot be seedbanked owing to inherent seed physiology or ecological factors. Specifically, we:

— determine how much genetic diversity has been captured in present-day *ex situ* collections—how well do *ex situ* collections, *as they are today*, represent the wild populations?
— resample the wild population genetic datasets to determine how much genetic diversity *could be* captured by varying collection sizes, assuming random sampling;
— calculate minimal collection size needed to capture 70% and 95% of the genetic diversity;
— test the influence of genus on the minimum collection needed to capture the genetic diversity and on a new statistic we term the 'genetic conservation gap'; and
— test other factors including species' genetic structure (reflecting fragmentation and gene flow) and allele frequency spectrum (reflecting demographic history).

## 2. Material and methods

### (a) Study species
We selected our target perennial trees and shrubs (table 1, see more information in the electronic supplementary material, table S1) because: (i) they are 'exceptional species' [13] that present difficulties for seedbanking (low seed production, low seed viability or recalcitrant seeds) and must be conserved *ex situ* as living plants; (ii) they cover a broad phylogenetic range (Monocots, Rosids, Magnoliids and Cycadophytes; see the electronic supplementary material), so our results can be applicable across plant groups; (iii) they represent variation in geographical ranges or numbers of populations; and (iv) all are threatened taxa, and some are critically endangered.

We studied taxa from five genera. Two subspecies of *Hibiscus waimeae*, subsp. *hannerae* and subsp. *waimeae* (Malvaceae), are endemic to the island of Kaua'i in Hawai'i. Both are threatened by habitat degradation, non-native herbivores, competition by invasive plants, seed predation by insects and a possible decline or loss of native pollinators. *Hibiscus waimeae* subsp. *hannerae* is rarer—approximately 200 adult individuals from three valleys remain and little or no regeneration occurs. *Magnolia ashei* (synonyms: *Magnolia macrophylla* subsp. *ashei* and *Magnolia macrophylla* var. *ashei*, Magnoliaceae) is a small deciduous tree with large leaves and flowers that occurs in only 10 counties in Florida (dataset presented in [19]). *Magnolia pyramidata* (synonym: *Magnolia fraseri* var. *pyramidata*) is a medium-sized deciduous tree reported from Texas to South Carolina, though with few occurrences in most states, except southern Alabama and northwestern Florida. *Pseudophoenix ekmanii* (Arecaceae) is restricted to a small area of the Dominican Republic and threatened by illegal destructive harvesting of sap. *Pseudophoenix sargentii*, the sister species of *P. ekmanii*, has the widest distribution of its genus, from Florida to Belize and east to Dominica, yet populations are isolated and locally imperiled. *Quercus georgiana* (Fagaceae) is a specialist on stone outcrops known from small, fragmented populations in Alabama, Georgia and South Carolina. *Quercus boyntonii* is a small tree occurring on sandstone outcrops in a few counties in Alabama, making it one of the most range-restricted United States oaks. *Quercus oglethorpensis*, formally described in 1940, occurs in sparse isolated populations across the southeast, also in granite outcrops. *Zamia decumbens* (Zamiaceae) is a slow-growing infrequently reproducing cycad from southern Belize, with less than 500 extant plants, while *Zamia lucayana* is a fast-growing, frequently coning cycad from the Bahamas, with one population of fewer than 1000 plants on a single island (here we expand on the dataset from [18]).

### (b) Sampling and genotyping
*In situ* populations were identified via Global Biodiversity Information Facility (GBIF) occurrence data and local collaborator networks. We attempted to sample as many wild populations as possible and to best represent the wild extant genetic diversity. *Ex situ* locations were identified via Botanic Gardens Conservation International's PlantSearch (https://members.bgci.org/data_tools/plantsearch), Beckman *et al.* [10] and personal contacts. Details of molecular markers, polymerase chain reaction (PCR) conditions, sampling and genotyping are presented in the electronic supplementary material. Briefly, leaf tissue was collected from *in situ* adult plants and *ex situ* adult and seedling plants. DNA was extracted using standard methodologies (CTAB or commercial kits), quantified and diluted. Approximately 10 microsatellites were amplified on each species using PCR and analysed using automated DNA sequencers such as ABI 3730. The software GENEIOUS was used to call alleles and export data to .csv files.

### (c) Computations
We performed two sets of calculations on the genetic datasets: how much genetic diversity *is currently* conserved in gardens, for each

species, and how much genetic diversity *could be* conserved with different collection sizes. All code and five datasets are available at: https://github.com/smhoban/IMLS_Safeguarding/tree/master and https://doi.org/10.5061/dryad.zgmsbcc74 [20].

The approach for determining how much genetic diversity *is currently* conserved was to: (i) genotype the *in situ* and *ex situ* individuals mentioned above, and (ii) calculate the percentage of *in situ* alleles (a metric of total genetic diversity) that are represented in the *ex situ* individuals (see [5,18]). This calculation of wild genetic diversity that is safeguarded *ex situ* is a measure of conservation success. We used the R package 'adegenet' [21] to tabulate all alleles. We then wrote R scripts to classify each allele into categories (see below), count each allele that was captured *ex situ* and then calculate the percentage of genetic diversity captured (number alleles captured divided by total number existing) for each allele category.

The approach for determining how much genetic diversity *could be* conserved under different strategies was to: (i) begin with the wild, *in situ* population genetic dataset; (ii) computationally resample this dataset for different numbers of individuals, representing different possible collection sizes that *could* have been used; (iii) for each resample, calculate the percentage of alleles in each of these potential *ex situ* collections, relative to the total *in situ* dataset; and (iv) assess the relationship between potential collection size versus genetic diversity captured. R scripts and a modified version of the 'sample.pop()' function from adegenet were used to create subsamples of the wild dataset ranging in size from two through to the total population size for each species. This procedure was repeated 75 000 times for each taxon.

To summarize the data, compare among taxa and inform a conservationist seeking to optimize their effort, we calculate a measure of 'sufficiency', defined as size needed for the collection to reach a minimum target percentage of genetic diversity (i.e. minimum sample size; this size is referred to as 'Ni' for the number of individuals in [22]). We calculated sufficiency to capture a minimum of 70% and 95% of genetic diversity. Mathematically, this is the first subsample size whose mean across 75 000 repetitions was greater than 70% or 95%. We use 95% as an arbitrary goal following most research in this topic (starting with [14]), and we use 70% based on Target 9 of the GSPC [23].

Sufficiency calculations will depend on the frequency of alleles the conservationist decides is 'important' [22,24]. Of course, rare alleles require much more sampling, on average, to capture. We therefore placed all alleles into five categories based on their frequency in each species: all alleles, very common (alleles greater than 0.10 frequency), common (alleles greater than 0.05 frequency), low frequency (frequency less than 0.10 and greater than 0.01), and rare (alleles less than 0.01 frequency). (The 'low frequency' category represents alleles in between common and rare alleles—rare enough to be hard to conserve, yet common enough to represent alleles under diversifying selection or adaptations to periodic pressure such as disease [22,24,25]). We use a regression to determine if collection size was a significant predictor of the proportion of alleles captured for each frequency class.

Lastly, the conservationist may decide that some very rare alleles are negligible. For example, alleles observed in only one or two individuals have a low chance of persisting. Moreover, some alleles that occur only once or twice in a dataset may be artefacts owing to genotyping error. We therefore tested two plausible assumptions—reduced dataset, where the conservationist is either not interested in extremely rare alleles or believes they may be artefacts, and full dataset, where the conservationist is interested in the value of every allele for its potential adaptive utility. In the reduced dataset, any alleles below a predefined threshold are excluded from consideration. We used a threshold of two alleles in the *in situ* dataset, which, for a population size of 200 diploid individuals, is a frequency of 0.005.

We tested the influence of 'genus' by performing three sets of ANOVAs, for each allele category, for full and reduced datasets, all with the predictor variable being 'genus' and the response variable being: (i) the amount of diversity currently conserved, (ii) sufficient size to reach 70% or 95% of alleles, and (iii) genetic conservation gap. An alternative explanatory factor might be the degree of population genetic structure, with theory predicting that more structured populations need, on average, larger collections [22,24]. Therefore, we performed linear regressions with the mean, minimum, maximum and standard deviation of pairwise FST (fixation index, a measure of population differentiation) among populations within a species as the predictor variable and the same three response variables just noted. A final explanatory factor might be species' allele frequency spectra which could reflect demographic history such as bottlenecks [26]. Therefore, we performed linear regressions with several spectra summaries (the proportion of alleles below 0.001, 0.05 or 0.01 frequency) as predictor variables and the same three response variables.

## 3. Results

For each question, we first present results for the reduced dataset, where alleles present in one or two copies are excluded from the analysis (see Material and methods), and then results for the full dataset, where all alleles are retained. Across all species, 21% of alleles were present in one or two copies.

### (a) Current genetic diversity of each taxon in *ex situ* collections

The percentage of *in situ* alleles currently present in *ex situ* collections is summarized in table 2. For the reduced dataset, only two species are conserved at ≥95%, *P. sargentii* and *Z. lucayana*. The percentage of genetic diversity safeguarded in collections is typically lower with the full dataset; only one species has ≥90% of genetic diversity protected overall (*Z. lucayana*) and none have ≥95%. Genus, population structure (FST) and allele frequency spectra were not a significant explanatory factor for *ex situ* conservation of any of the allele categories after Benjamani and Hochberg multiple comparison correction. However, the number of plants in current collections is a good predictor for the proportion of low frequency alleles and all alleles, showing a logarithmic relationship (figure 1).

### (b) Sufficiency- minimum needed to capture a threshold per cent of diversity

By resampling the *in situ* datasets for a number from two to the total number of individuals, we calculated genetic diversity that could be captured for each potential collection size under optimal sampling, and we then plotted genetic diversity against collection size (figure 2). Comparing the panels demonstrates the importance of deciding whether to include these extremely rare alleles when calculating the minimum size. The relative 'order' of the species, however, does not change much. In both cases, the minimum size to reach a threshold of 95% (a sufficiency measure) differs among species with no clear taxonomic signal (with the exception of oaks, which cluster together). Depending on the species and dataset, sufficient size to capture 95% of wild genetic diversity ranges from 24 to 207 individuals. Sufficiency to reach a threshold of 70% is shown in the electronic supplementary material. Neither genus nor population structure were significant predictors of minimum collection size for any allele category, for any threshold, for full or reduced datasets. However, the allele frequency

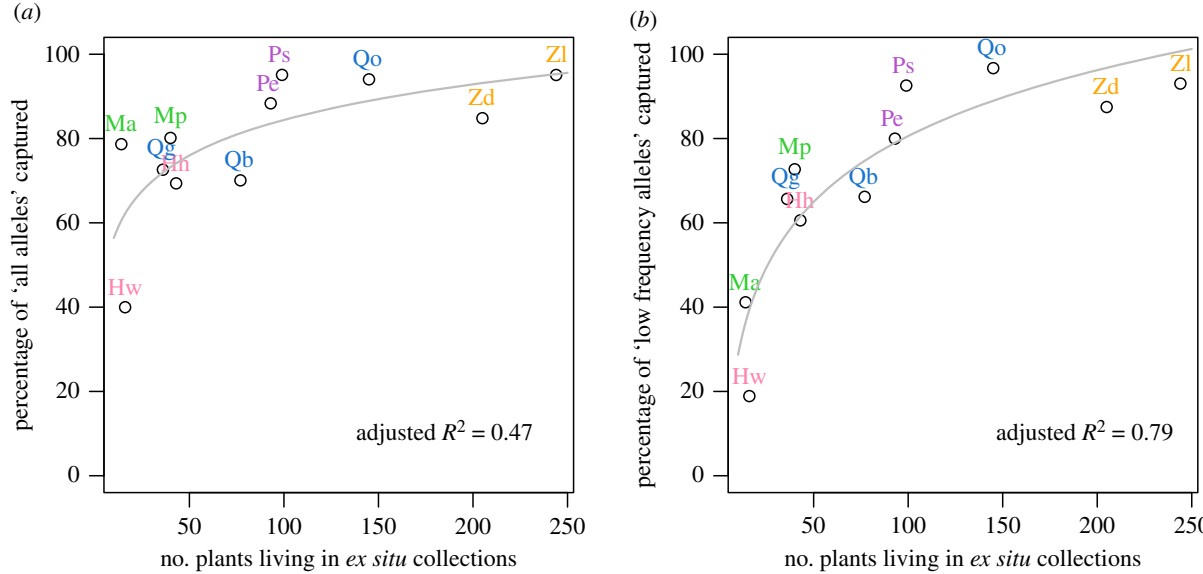

**Figure 1.** Collection size (x-axis) in relation to allele capture (y-axis), for two types of alleles: all alleles (a) and low frequency alleles (b). The reduced dataset was used for both plots. Grey lines represent model fit using log(number of plants), with adjusted $R^2$ shown. (Online version in colour.)

**Table 2.** In situ and ex situ collection size and percentage of genetic diversity currently preserved in five allele frequency categories. (n in situ and n ex situ refer to the number of plants genotyped. Parentheses show calculations under the full dataset (i.e. including every allele, for comparison to the reduced dataset, which excludes alleles found in only one or two copies).)

| taxon | n in situ | n ex situ | all_reduced (all_full) | very common | common | low frequency | rare_reduced (rare_full) |
|---|---|---|---|---|---|---|---|
| | number of plants genotyped in this study | | percentage (%) | | | | |
| H. w. hannerae | 157 | 43 | 69 (53) | 100 | 93 | 61 | 00 (13) |
| H. w. waimeae | 73 | 16 | 40 (28) | 90 | 59 | 19 | NA (00) |
| P. ekmanii | 201 | 93 | 88 (67) | 100 | 96 | 80 | 50 (06) |
| P. sargentii | 122 | 99 | 95 (79) | 100 | 98 | 93 | NA (37) |
| M. ashei | 104 | 14 | 79 (69) | 93 | 92 | 41 | NA (18) |
| M. pyramidata | 113 | 40 | 80 (68) | 100 | 97 | 73 | NA (35) |
| Q. boyntonii | 244 | 77 | 70 (60) | 100 | 100 | 66 | 29 (32) |
| Q. georgiana | 223 | 36 | 73 (65) | 100 | 92 | 66 | 39 (33) |
| Q. oglethorpensis | 187 | 145 | 94 (78) | 100 | 100 | 97 | 67 (37) |
| Z. decumbens | 374 | 205 | 85 (77) | 100 | 100 | 88 | 22 (26) |
| Z. lucayana | 120 | 244 | 95 (91) | 100 | 100 | 93 | NA (63) |
| mean | 174 | 81 | 76 (65) | 98 | 92 | 67 | 18 (26) |
| s.d. | 85 | 77 | 16 (16) | 3.8 | 12 | 23 | 15 (17) |

spectra were partially explanatory—the proportion of alleles below 0.05 frequency was a good predictor of minimum collection size for all alleles and common alleles, for full and reduced datasets, but only for meeting a threshold of 70% of alleles, not for 95% (see the electronic supplementary materials).

## 4. Discussion

A first key finding is that *there is wide variation in the percentage of genetic diversity currently safeguarded in* ex situ *collections of threatened woody plants, and most* ex situ *collections do not sufficiently*

*capture the genetic diversity of wild populations.* In our study, the genetic diversity (per cent of *in situ* alleles) captured in existing *ex situ* collections varies from 40% to 95% for the reduced dataset (an optimistic assumption, excluding the rarest alleles from analysis), and from 28% to 91% for the full dataset (a conservative assumption, retaining all alleles). On average, larger collections capture more genetic diversity, though size is not a perfect predictor of diversity captured. Even within genera, some taxa were collected more 'efficiently' than others: for example, *Q. georgiana* safeguards equivalent diversity to *Q. boyntonii* in less than half the collection size (figure 1). Our work shows that it is possible to calculate progress towards global

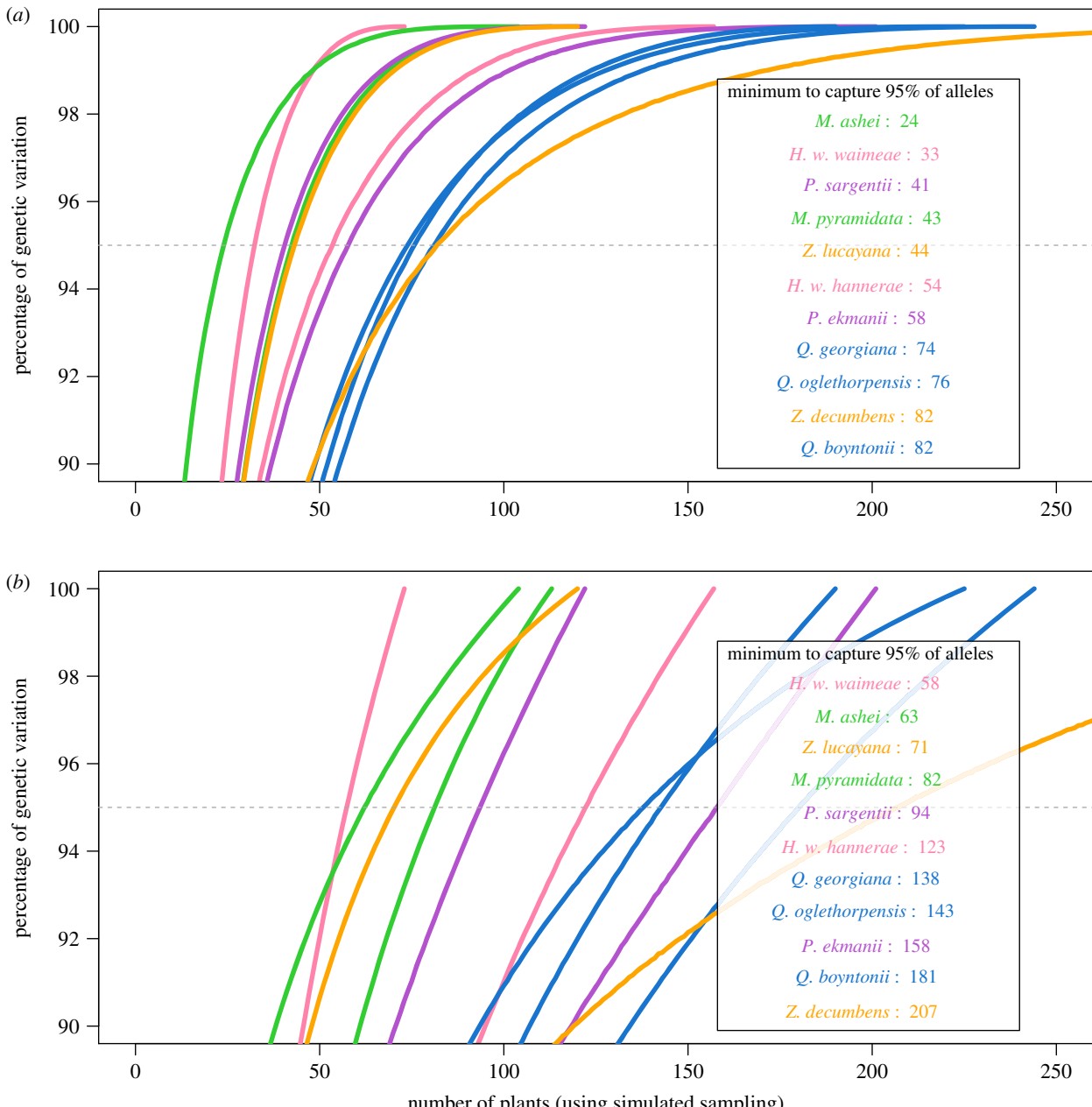

**Figure 2.** Percentage of genetic variation (*y*-axis) captured relative to the number of plants randomly sampled (*x*-axis), shown for (*a*) reduced dataset and (*b*) full dataset. The *y*-axis starts at 90% to focus on a 95% goal. Each genus is coded by colour. The legend shows the number of plants needed to get 95% of the alleles (where the taxon curve crosses the dashed line), ordered from taxon with the fewest number of plants needed. If a similar size was required for each genus, they would be adjacent in this ordered list (which is not observed, except for *Quercus*). (Online version in colour.)

conservation targets such as the GSPC and the Convention on Biological Diversity with genetic data (though this assessment requires substantial resources—approx. 10 000 USD per species for genotyping). For the reduced dataset, only three of 11 species have 95% of alleles conserved (notably with substantially different collection sizes, 99, 145 and 244 individuals, table 2), while all but two (the *Hibiscus* taxa) have at least 70% of alleles conserved (satisfying Target 9 of the GSPC). However, using the full dataset, no taxa achieve 95% and only three achieve 70%. Of course, the percentage captured depends on the allele category examined (table 1). For example, for common alleles, six taxa meet the 95% threshold and all 11 meet the 70% threshold. We predict that, unfortunately, most threatened species in botanic gardens are conserved to an even lesser degree than we find, as our target taxa have been the focus of intensive conservation efforts.

For the main question, 'should taxa within a genus, which tend to share many biological traits relevant to genetic diversity and structure, have similar collection sizes for safeguarding genetic diversity *ex situ*?', our results suggest that the answer is 'no': *collectors cannot assume that the same protocol will work equally for all taxa in a genus* (see also [18]). Genus did not explain significant variation in any of our quantitative results—in fact, taxa within a genus required substantially different strategies to capture equivalent levels of genetic diversity. For example, for a goal of 95% of alleles, one *Zamia* species requires almost double the size (91% more) of the other. We also did not find that population structure affected genetic diversity captured in *ex situ* collections or sample size needed to preserve target diversity. It is likely that biogeographical and demographic history, traits such as the frequency of reproduction and sampling strategy affect ideal sampling for genetic

**Table 3.** Genetic conservation gap—proportion of current collection size that could conserve the same percentage of alleles (left) and potential proportional increase in genetic diversity capture while using the same collection size (right), if random sampling could be performed.

| | proportion of current collection size | | potential increase in genetic diversity capture | |
|---|---|---|---|---|
| | full dataset | reduced dataset | full dataset | reduced dataset |
| H. hannerae | 0.28 | 0.26 | 1.43 | 1.34 |
| H. waimeae | 0.19 | 0.19 | 2.41 | 2.07 |
| M. ashei | 0.50 | 0.43 | 1.15 | 1.15 |
| M. pyramidata | 0.53 | 0.48 | 1.21 | 1.18 |
| P. ekmanii | 0.26 | 0.35 | 1.30 | 1.12 |
| P. sargentii | 0.39 | 0.41 | 1.21 | 1.05 |
| Q. boyntonii | 0.35 | 0.27 | 1.34 | 1.35 |
| Q. georgiana | 0.58 | 0.50 | 1.16 | 1.18 |
| Q. oglethorpensis | 0.39 | 0.48 | 1.22 | 1.30 |
| Z. decumbens | 0.19 | 0.16 | 1.22 | 1.17 |
| Z. lucayana | 0.21 | 0.18 | NA | NA |
| mean | 0.35 | 0.34 | 1.37 | 1.29 |

conservation and require a further study. *Quercus* was the one exception. The minimum size for all three oak species is about 80 (for the reduced dataset). All three are rare, wind-pollinated, habitat specialists, but they have different geographical distributions (from a few countries to several states). Interestingly, sample size to sufficiently capture genetic diversity partially corresponds to allele frequency profiles, which may reflect demographic history, e.g. bottlenecked populations.

While there is not a general 'best' strategy to apply to a particular genus (excepting *Quercus*), *the taxonomic breadth of our study illustrates the 'bounds' of minimum size for threatened plants.* On the optimistic side, results for the reduced dataset suggest that many taxa will require collections from 25 to 82 individuals to capture 95% of all alleles; more conservatively, results for the full dataset suggest 58–207 individuals. The range is dependent on the decision to include or exclude ultra-rare alleles. While some of these alleles may be genotyping errors, probably most are not; genotyping errors in microsatellites are typically 1–5%, while these alleles represent 21% of our observed alleles. In general, *if nothing is known about a species' biology, demography or genetics, the sufficient minimum collection size will usually fall between approximately 30 and 200 plants if sampling randomly* (though see Caveats below). This is consistent with a recent modelling study [22], suggesting bounds of approximately 100–300, and with suggestions by Hoban & Strand [15].

Comparing the potential genetic capture of a random sample (figure 2) with the actual genetic capture in *ex situ* collections today (table 1) identifies a critical observation: *current* ex situ *collections are suboptimal and could be more efficient and/or effective.* We call this difference between actual and optimal sampling the 'genetic conservation gap'. This gap can refer to either: the proportion of current collection size which could achieve the same conservation outcome or the proportional increase in genetic capture possible using the same number of plants. For example, *Z. lucayana* collections capture 95% of alleles in 244 samples, while a more optimal field sampling could capture the same genetic diversity in only 44 samples, almost six times smaller. Our results suggest that *ex situ*

*collections could be one half to one sixth the current size if more optimally collected* within and among populations across a species geographical range (table 3). Alternatively, *ex situ collections could remain the same size and harbour much more genetic diversity* under optimum sampling of wild germplasm. For example, *Q. boyntonii* collections (77 individuals) preserve 70% of diversity (table 2), but random sampling could preserve 94% of diversity, a remarkable increase in conservation success for the same collection size (table 3).

The limitations and biases of real-world seed collections have been previously discussed, but our results may be the first to quantify how much better random sampling will perform relative to the diversity in collections now. This gap is not justification to reduce the size of collections; we use it to highlight that collections could be more optimally structured. The reasons for suboptimal gene conservation success may include: difficulty in visiting all wild populations and/or making systematic sampling within populations; seed collectors rarely sample over multiple years; artificial selection during seed cleaning and cultivation [27]; and curators often share seed or plant cuttings among gardens rather than making new acquisitions from wild populations [5,28,29]. We also note that our calculations are minimum sampling to get one copy of each allele or genetic variant, which does not ensure duplication or 'backup' to protect against disasters or natural attrition [30]. In most cases, a conservationist would desire multiple copies; Hoban [22] demonstrated how to calculate sufficient sample size for duplication and recommend at least five copies for backup. The degree of duplication of alleles in current collections should be assessed.

We know of no other study analysing and optimizing *ex situ* genetic diversity across multiple genera in plants. However, Whitlock *et al.* [31] estimated how many populations should be protected to preserve genetic diversity *in situ*, in eight 'widespread but declining' species; they did not examine *ex situ* collections. They sampled 16–42 populations per species, genotyped them and subsampled by the number of populations (assuming that all individuals were protected *in situ*). Capturing 70% of rare alleles necessitated protecting

5–15 populations per species *in situ*. Their study differs from ours in that they studied widespread species, and resampled the number of populations rather than the number of individuals. Nonetheless, they demonstrated little relationship to the mating system, similar to our findings.

## (a) Caveats

Two key differences exist between the computational resampling which we used in this study and real-world plant conservation: sampling strategy and collection attrition. These two points mean that *most real-world collections will capture significantly less genetic diversity than shown in the tables and plots in this paper—what we present are absolute minimum sample sizes.*

First, the resampling algorithm chooses individuals randomly and visits all wild populations—an ideal scenario rarely reached in reality. If non-random sampling is used, Hoban & Strand [15] suggest sampling roughly twice as many seeds (see also [29]). Also, in contrast to simulated sampling, collectors typically sample from a few populations and dozens to hundreds of seeds per plant rather than one. Previous work shows that sampling multiple seeds per plant will capture less genetic diversity than sampling fewer seeds per plant but including more parent plants [15,24,25]. There is no known formula to calculate how much genetic diversity is reduced when sampling multiple seeds on a plant, but some examples can be found in Kashimshetty *et al.* [29] and Hoban *et al.* [25].

Second, our calculations assume that no plants in the collection die over time. However, in reality, some seeds fail to germinate, and living plants experience disease, damage and senescence [4,30]. If germination and mortality rates are established [32,33], the amount to collect from the wild should be increased to ensure that even after losses, the minimum number of plants will remain living (see [22]). For example, if a minimum collection size is 82, and only 10% of seeds survive to adulthood, then 820 seeds should be collected. Thus, *most collections should be larger than the values shown in our figures and tables, probably multiple times larger (e.g. 150–1000 total individuals).*

There are several considerations beyond the scope of this study. First, we considered rare, perennial woody plant species, and our results probably do not apply to all species. Collections for common species may need tens of thousands to millions of seeds [25,28], though more work is needed to establish how to collect local adaptations and useful traits such as disease resistance. Also, 11 species, while a larger study than any previously undertaken, are still a moderate amount for testing the influence of population structure and species' traits. Additional genetic studies are recommended to further test our findings. The DNA markers used here, microsatellites, typically represent neutral diversity, not adaptive, and are not fully representative of the whole genome. Also, although we used multiple methods to find as many known populations as possible (herbaria vouchers and personal contacts) for our sampling, it is likely that some wild populations and/or individuals were unsampled in our study, which would increase minimum sizes for rare alleles, but our overall conclusions would be unchanged. Finally, *ex situ* collection management necessitates further study [34], including how institutions coordinate collection efforts [35,36], how repeated sampling over time can capture adaptive change and how to captive breed plant collections [12,16].

## (b) Recommendations and conclusion

Our study of 11 threatened, perennial taxa demonstrates clearly that most *ex situ* collections are insufficiently safeguarding genetic diversity, and that the appropriate collection size needed to sufficiently and efficiently capture genetic diversity in *ex situ* collections is unlikely to be similar for taxa in the same genus, with the possible exception of *Quercus*. Nonetheless, optimal size across all genera is within an order of magnitude (typically 30–200). More case studies in other genera are needed to determine the relative importance of each factor (see also [22]). We also found that collections could capture 40% greater diversity in the same number of plants, or could retain current genetic diversity levels with many fewer trees by closing the genetic conservation gap. We recommend that curators make optimal use of limited garden space by, when possible, using genetic data to calculate how many specimens are needed to capture the desired level of genetic diversity. If extensive genetic work is infeasible, we recommend to maximize genetically unique samples by adhering to ideal sampling practices: few seeds per plant but many unique maternal plants and visit most or all wild populations. Our results build on previous work, showing that preserving species *ex situ* will take more resources than currently committed. To accommodate additional plants, curators must make hard decisions to deaccession (i.e. remove) material that has a low conservation value (numerous clones, large numbers of full siblings, accessions that are well represented in many other gardens and very commonly grown species). Of course, redundancy is needed, with some duplicates maintained through the metacollection. Lastly, we identified a heretofore overlooked factor that may be more influential than species biology—decisions about the level and type of genetic diversity to preserve (i.e. the minimum allele frequency threshold a collection must capture, and the category of allele prioritized). Discussion and resolution on these questions, by the *ex situ* community, is needed for the quantitative optimization of *ex situ* collections.

Data accessibility. Additional methods including details on each species, and additional results as noted in the paper, are in the electronic supplementary PDF. The code for resampling, plus a readme file explaining how to run the code, and five species data files are included in a GitHub repository for which a link is included in the manuscript: https://github.com/smhoban/IMLS_Safeguarding/tree/master and from the Dryad Digital Repository: https://doi.org/10.5061/dryad.zgmsbcc74 [20].

Authors' contributions. All authors conceived and planned the study. Major data collection was by S.D., J.F., O.G., P.K., A.T.K., A.W.M., M.P., V.S., E.S., R.T., S.W., and M.P.G. S.H. drafted the paper and performed statistical analysis. All authors contributed to writing the paper.

Competing interests. We declare we have no competing interests

Funding. This project was made possible through support of the Institute of Museum and Library Services (grant nos MA-05-12-0336-12, MA-30-14-0123-14, MA-30-18-0273-18 and MG-30-16-0085-16) and the National Science of Foundation (grant nos DEB 1050340, DBI 1203242 and DBI 1561346). Fieldwork was supported by the Plant Exploration Fund, Association of Zoological Horticulture, SOS—Save Our Species (grant no. 2012A-035) and Mohamed bin Zayed Species Conservation Fund (projects 0925331, 12254271 and 162512606).

Acknowledgement. We thank the 63 botanic gardens who contributed plant tissue to this project, who are listed in the electronic supplementary material, E. Schumacher for coding assistance, C. Johnson for manuscript preparation, and numerous volunteers in the field.

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
