## [Reviewer comments · Proceedings of the Royal Society B: Biological Sciences]

Review History

RSPB-2020-0102.R0 (Original submission)

Review form: Reviewer 1

Recommendation

Accept with minor revision (please list in comments)

Scientific importance: Is the manuscript an original and important contribution to its field?

Excellent

General interest: Is the paper of sufficient general interest?

Good

Quality of the paper: Is the overall quality of the paper suitable?

Excellent

Is the length of the paper justified?

Yes

Should the paper be seen by a specialist statistical reviewer?

No

Do you have any concerns about statistical analyses in this paper? If so, please specify them explicitly in your report.

No

It is a condition of publication that authors make their supporting data, code and materials available - either as supplementary material or hosted in an external repository. Please rate, if applicable, the supporting data on the following criteria.

Is it accessible?

Yes

Is it clear?

Yes

Is it adequate?

Yes

Do you have any ethical concerns with this paper?

No

Comments to the Author

This well written article furthers the science related to producing generalized guidance for sample sizes of taxa for ex situ conservation, working across multiple genera. It is elegantly designed, particularly with reference to standard literature (ie. 95% of genetic diversity - Marshall and Brown) as well as current global targets (ie. 70% from the GSPC), and the binning/categories of common to rare alleles. Its main finding is that generalization by genus is not possible, although more general recommendations can be made. I provide a few comments below to improve the draft. I do think the article - including possibly the title - should be revised to qualify it further with regard to its specific focus (on recalcitrant, threatened woody perennials).

Lines:

16-21- probably need references for these international agreements

23- Khoury et al. 2019 assess 3% for ex situ; 40% in situ; and almost 3% in combination. You might also note that this research did not measure genetic diversity directly but rather used geographic methods as proxy

26- although this article focuses on wild plants, I think it is worth noting here the very significant contribution of seedbanks (or genebanks) to genetic diversity conservation for species relevant to agrobiodiversity, particularly domesticated species.

32 I think that 'cannot' is too strong a word. Research increasingly shows, indeed, that most recalcitrant species can in fact be conserved given adequate research. The issue is more one of resource mobilization and priorities. Please consider an alternative like "are very challenging" or something like that. Note at 36 that its likely that cryo is an option for almost all species; again just a matter of resources and research.

77 and 96-102 - given the broad claim here, it will be extremely important to note the caveats and constraints that may not make the findings generalizable. Among potential others, these include: focus on perennial trees and shrubs (rather than across life forms in plants), focus on recalcitrant, the small (species) sample size of the study, etc. An alternative is to walk back the broad claims in the introduction and make it clearer what you are focusing on in the article (including in the title), e.g. on threatened woody perennials.

162 why 75k?

191 collector or collection?

220 no need to capitalize Ex

Table 2- just a few questions for clarity- did you genotype every ex situ and every in situ plant noted in the table? And how certain are you that you captured all in situ plants/populations?

276- yes! But with very substantial resources and effort! Should be noted.

292 you might consider making this the title of the article, i.e. something like "Taxonomic similarity does not predict necessary sample size for genetic conservation (for threatened perennials- see earlier note)"

332 would it be possible to compute a third metric- the amount that collection size could go up or down in terms of sample size, to achieve ideal targets (70% or 95% or whatever)? This seems even more pertinent than the two metrics you currently provide.

356 yes and there are advantages to having more than one copy (i.e. a backup). It might be nice to note this more clearly and even recommend what a 'safe' genetic conservation strategy might look like (i.e. maybe 44 samples could conserve *Z. lucayana*, but perhaps each should be at least duplicated once (i.e. in two botanic gardens or something like that). At lines 429-437 same point; you might note that duplication could happen rather than sampling more plants.

448- here is where you might mention that woody perennials (and recalcitrant ones at that) might not reflect overall plant diversity needs, and also the relatively small sample size of the study in terms of species/genera (even if its much larger than anything done before).

Review form: Reviewer 2 (Mitchell McGlaughlin)

Recommendation

Accept with minor revision (please list in comments)

Scientific importance: Is the manuscript an original and important contribution to its field?

Excellent

General interest: Is the paper of sufficient general interest?

Good

Quality of the paper: Is the overall quality of the paper suitable?

Excellent

Is the length of the paper justified?

Yes

Should the paper be seen by a specialist statistical reviewer?

No

Do you have any concerns about statistical analyses in this paper? If so, please specify them explicitly in your report.

No

It is a condition of publication that authors make their supporting data, code and materials available - either as supplementary material or hosted in an external repository. Please rate, if applicable, the supporting data on the following criteria.

Is it accessible?

Yes

Is it clear?

Yes

Is it adequate?

Yes

Do you have any ethical concerns with this paper?

No

Comments to the Author

This work represents a substantial contribution to our thinking about diversity in ex situ plant collections. The sampling and analyses are appropriate for your question and your results are clearly presented. I have provided some line specific comments below and I would recommend some minor additions/expansions in two areas:

- 1) Readers would benefit from more recommendations on what next steps should be based on your data. Would you recommend replacing/removing some collections from gardens that are redundant? Should there be more guidelines for how collections are made? Do you recommend more genetic work to better understand the drivers of diversity? Your results open up a lot of additional questions only some of which are addressed in the discussion.
- 2) Making sure you are not overstating your results, specifically as it relates to organismal characteristics, range, collection effort. The conclusions you can draw related to necessary ex situ collection sizes to reach specific levels of diversity are very robust, but you lack insight into the drivers of wild diversity.

Specific Comments

Ln 87 – It is unclear what is meant by ‘threshold’. In later sections you make clear what thresholds you are examining, but it is unclear here.

Ln 104-124 – Please add family after each taxon.

Ln 157 – ‘possible sizes’ needs an additional description like ‘collection’

Ln 272 – I count 6 species exceeding 70%

Ln 304-305 – It would be useful to provide more details related to demographic history. Did you explore potential bottlenecks? You have not provided any specific information about genetic diversity, so it is difficult to know the role it plays.

Ln 306 – ‘minimum collection size’

Ln 442 – This is an area where making more concrete recommendations would be useful.

Ln 454 – Slightly more details on why you feel confident that most populations are represented in your dataset would be useful.

Ln 468-470 – This is somewhat overstated. I don’t disagree with the listed traits being important, but you did not explicitly test any of them (Fst gets at some indirectly). You hypothesize that the identified traits/characteristics are important, but beyond genus not being a predictor you do not specifically test any of these.

Decision letter (RSPB-2020-0102.R0)

19-Feb-2020

Dear Dr Hoban:

Your manuscript has now been peer reviewed and the reviews have been assessed by an Associate Editor. The reviewers’ comments (not including confidential comments to the Editor) and the comments from the Associate Editor are included at the end of this email for your reference. As you will see, the reviewers and the Editors have raised some concerns with your manuscript and we would like to invite you to revise your manuscript to address them.

Research ethics:

Use of animals and field studies:

Please submit a copy of your revised paper within three weeks. If we do not hear from you within this time your manuscript will be rejected. If you are unable to meet this deadline please let us know as soon as possible, as we may be able to grant a short extension.

Best wishes,
Professor Gary Carvalho
mailto:proceedingsb@royalsociety.org

Associate Editor
Comments to Author:

I have now received two reviews of the manuscript "The optimal size of an ex situ conservation population: a comparison among 11 taxa in 5 genera." Both reviewers thought that the manuscript would make a significant contribution and was of general interest; I agree.

Each of the reviewers has modest, but important, suggestions for changes that would strengthen the manuscript. I carefully read the manuscript and spent some time looking over the supplemental information. Based on my reading, I have identified additional areas where the manuscript could be strengthened. Based on the positive reviews, I request a revision that addresses each of the concerns raised.

Associate Editor Comments:

1) Where are the genetic results? Please provide a summary table with the results from the genetic analyses in the text or the supplemental material. The text reports second order results that come from the genetic analyses. The primary results need to be reported.

2) At slightly over 6 pages, about 50% of the manuscript, the Discussion is too long. Throughout, the length can be reduced, try for 1-2 pages. In particular, lines 385-407 and 411-460 seem like they could be cut a good bit and the conclusion is redundant with previous statements.

3) Table 3.

a) Elsewhere in the manuscript reduced and full data set results are reported. Please do that here as well.

b) The first column, "x times smaller" is awkward – people don't usually use this phrasing. Recommend instead reporting the proportion of the current size that would meet the same result – this will resonate more easily with a reader.

4) line 130-131. Please provide a general summary of how the sampling and acquiring the genetic material were conducted in the text so the reader does not need to go to the supplemental section to understand how the research was conducted.

5) line 145ff. This summary might have been more helpful at the beginning of the methods. The first 3 numbers in the list occur prior to the “computations” section that the list is placed in. Or else, perhaps it isn’t really needed.

6) line 171. Citation?

7) line 178. Common alleles are >0.05 and low frequency alleles are between $0.01 < \text{low-freq} < 0.1$. So, between 0.05 and 0.1 alleles are both common and low frequency?? This seems awkward, suggest rethinking categories.

8) line 277. What are GSPC and CBD?

9) Supplemental Material: I looked through the supplemental material a number of times. It is very difficult to assess what was done. I recommend reorganizing it to put all sampling together, all methods together, etc. This would also yield more consistency in the reporting for each of the taxa.

10) Supplemental Material: provide exact locations of sampled wild material.

11) Supplemental Material: Read over the main Supplemental Material file with keen editorial eye. For example, remove word comments, at one point I noticed the text said “I” and not “we”, try for consistency across the presentation of the different species.

Reviewer(s)' Comments to Author:

Referee: 1

Comments to the Author(s)

This well written article furthers the science related to producing generalized guidance for sample sizes of taxa for ex situ conservation, working across multiple genera. It is elegantly designed, particularly with reference to standard literature (ie. 95% of genetic diversity – Marshall and Brown) as well as current global targets (ie. 70% from the GSPC), and the binning/categories of common to rare alleles. Its main finding is that generalization by genus is not possible, although more general recommendations can be made. I provide a few comments below to improve the draft. I do think the article – including possibly the title – should be revised to qualify it further with regard to its specific focus (on recalcitrant, threatened woody perennials).

Lines:

16-21- probably need references for these international agreements

23- Khoury et al. 2019 assess 3% for ex situ; 40% in situ; and almost 3% in combination. You might also note that this research did not measure genetic diversity directly but rather used geographic methods as proxy

26- although this article focuses on wild plants, I think it is worth noting here the very significant contribution of seedbanks (or genebanks) to genetic diversity conservation for species relevant to agrobiodiversity, particularly domesticated species.

32 I think that ‘cannot’ is too strong a word. Research increasingly shows, indeed, that most recalcitrant species can in fact be conserved given adequate research. The issue is more one of resource mobilization and priorities. Please consider an alternative like “are very challenging” or something like that. Note at 36 that its likely that cryo is an option for almost all species; again just a matter of resources and research.

77 and 96-102 - given the broad claim here, it will be extremely important to note the caveats and constraints that may not make the findings generalizable. Among potential others, these include: focus on perennial trees and shrubs (rather than across life forms in plants), focus on recalcitrant, the small (species) sample size of the study, etc. An alternative is to walk back the broad claims in the introduction and make it clearer what you are focusing on in the article (including in the title), e.g. on threatened woody perennials.

162 why 75k?

191 collector or collection?

220 no need to capitalize Ex

Table 2- just a few questions for clarity- did you genotype every ex situ and every in situ plant noted in the table? And how certain are you that you captured all in situ plants/populations?

276- yes! But with very substantial resources and effort! Should be noted.

292 you might consider making this the title of the article, i.e. something like "Taxonomic similarity does not predict necessary sample size for genetic conservation (for threatened perennials- see earlier note)"

332 would it be possible to compute a third metric- the amount that collection size could go up or down in terms of sample size, to achieve ideal targets (70% or 95% or whatever)? This seems even more pertinent than the two metrics you currently provide.

356 yes and there are advantages to having more than one copy (i.e. a backup). It might be nice to note this more clearly and even recommend what a 'safe' genetic conservation strategy might look like (i.e. maybe 44 samples could conserve *Z lucayana*, but perhaps each should be at least duplicated once (i.e. in two botanic gardens or something like that). At lines 429-437 same point; you might note that duplication could happen rather than sampling more plants.

448- here is where you might mention that woody perennials (and recalcitrant ones at that) might not reflect overall plant diversity needs, and also the relatively small sample size of the study in terms of species/genera (even if its much larger than anything done before).

Referee: 2

Comments to the Author(s)

This work represents a substantial contribution to our thinking about diversity in ex situ plant collections. The sampling and analyses are appropriate for your question and your results are clearly presented. I have provided some line specific comments below and I would recommend some minor additions/expansions in two areas:

- 1) Readers would benefit from more recommendations on what next steps should be based on your data. Would you recommend replacing/removing some collections from gardens that are redundant? Should there be more guidelines for how collections are made? Do you recommend more genetic work to better understand the drivers of diversity? Your results open up a lot of additional questions only some of which are addressed in the discussion.
- 2) Making sure you are not overstating your results, specifically as it relates to organismal characteristics, range, collection effort. The conclusions you can draw related to necessary ex situ collection sizes to reach specific levels of diversity are very robust, but you lack insight into the drivers of wild diversity.

Specific Comments

Ln 87 - It is unclear what is meant by 'threshold'. In later sections you make clear what thresholds you are examining, but it is unclear here.

Ln 104-124 - Please add family after each taxon.

Ln 157 - 'possible sizes' needs an additional description like 'collection'

Ln 272 - I count 6 species exceeding 70%

Ln 304-305 - It would be useful to provide more details related to demographic history. Did you explore potential bottlenecks? You have not provided any specific information about genetic diversity, so it is difficult to know the role it plays.

Ln 306 - 'minimum collection size'

Ln 442 – This is an area where making more concrete recommendations would be useful.

Ln 454 – Slightly more details on why you feel confident that most populations are represented in your dataset would be useful.

Ln 468-470 – This is somewhat overstated. I don't disagree with the listed traits being important, but you did not explicitly test any of them (Fst gets at some indirectly). You hypothesize that the identified traits/characteristics are important, but beyond genus not being a predictor you do not specifically test any of these.

Author's Response to Decision Letter for (RSPB-2020-0102.R0)

See Appendix A.

Decision letter (RSPB-2020-0102.R1)

26-Mar-2020

Dear Dr Hoban

I am pleased to inform you that your Review manuscript RSPB-2020-0102.R1 entitled "Taxonomic similarity does not predict necessary sample size for ex situ conservation: a comparison among five genera" has been accepted for publication in Proceedings B.

The referee(s) do not recommend any further changes. Therefore, please proof-read your manuscript carefully and upload your final files for publication. Because the schedule for publication is very tight, it is a condition of publication that you submit the revised version of your manuscript within 7 days. If you do not think you will be able to meet this date please let me know immediately.

To upload your manuscript, log into <http://mc.manuscriptcentral.com/prsb> and enter your Author Centre, where you will find your manuscript title listed under "Manuscripts with Decisions." Under "Actions," click on "Create a Revision." Your manuscript number has been appended to denote a revision.

You will be unable to make your revisions on the originally submitted version of the manuscript. Instead, upload a new version through your Author Centre.

- 1) A text file of the manuscript (doc, txt, rtf or tex), including the references, tables (including captions) and figure captions. Please remove any tracked changes from the text before submission. PDF files are not an accepted format for the "Main Document".
- 2) A separate electronic file of each figure (tiff, EPS or print-quality PDF preferred). The format should be produced directly from original creation package, or original software format. Please note that PowerPoint files are not accepted.
- 3) Electronic supplementary material: this should be contained in a separate file from the main text and the file name should contain the author's name and journal name, e.g. `authorname_procb_ESM_figures.pdf`

All supplementary materials accompanying an accepted article will be treated as in their final form. They will be published alongside the paper on the journal website and posted on the online figshare repository. Files on figshare will be made available approximately one week before the accompanying article so that the supplementary material can be attributed a unique DOI. Please see: <https://royalsociety.org/journals/authors/author-guidelines/>

4) Data-Sharing and data citation

It is a condition of publication that data supporting your paper are made available. Data should be made available either in the electronic supplementary material or through an appropriate repository. Details of how to access data should be included in your paper. Please see <https://royalsociety.org/journals/ethics-policies/data-sharing-mining/> for more details.

If you wish to submit your data to Dryad (<http://datadryad.org/>) and have not already done so you can submit your data via this link <http://datadryad.org/submit?journalID=RSPB&manu=RSPB-2020-0102.R1> which will take you to your unique entry in the Dryad repository.

Once again, thank you for submitting your manuscript to Proceedings B and I look forward to receiving your final version. If you have any questions at all, please do not hesitate to get in touch.

Sincerely,

Professor Gary Carvalho
Editor, Proceedings B
<mailto:proceedingsb@royalsociety.org>

Associate Editor Board Member
Comments to Author:

Thank you for the work that you have put into revising this manuscript – the revision is thorough and thoughtful.

I have only one further request to enhance readability. The reorganization of the Supplemental Material definitely makes it easier to read and extract information from. However, could you also take the tables for the sampling and primers for each species and either put them at the end of the document or place them in a separate excel file? Then, the methods can be easily read through. Minor detail – this is a nice piece of work!

Decision letter (RSPB-2020-0102.R2)

03-Apr-2020

Dear Dr Hoban

I am pleased to inform you that your manuscript entitled "Taxonomic similarity does not predict

necessary sample size for ex situ conservation: a comparison among five genera" has been accepted for publication in Proceedings B.

Your article has been estimated as being 8 pages long. Our Production Office will be able to confirm the exact length at proof stage.

Open Access

Paper charges

Sincerely,

Proceedings B

Appendix A

>>Our responses are in bold and preceded by “>>” All lines number refer to the revised manuscript in Word format with Track Changes on, with Changes “in line” (using crossouts), *not* using in “bubbles” in the side margin (which affects line numbers). They should be correct in the uploaded PDF.

>>Thank you for the opportunity to revise the paper. We appreciate the thoughtful suggestions and positive comments, and we think we have addressed all identified issues. These include: substantial decrease in length of discussion, addition of basic summary statistics to the Supplemental, changes to presentation of Table 3, reorganization of Supplemental, addition of geospatial location of populations for species not threatened by poaching, a change in title, clarification to several methodological questions, further clarity on what kind of species the results apply to and how far results can be extrapolated, and a couple additions on next steps and recommendations. One suggestion we did not implement- reviewer 1 asked if a particular extension of our methodology is possible (calculating how to build on a current collection rather than our approach- calculating how to build a new collection), and while we respond that it is possible, this would be an additional paper’s worth of method development and we save that for a future paper. Aside from this, we agree to all suggestions from the Editor and reviewers. In addition note that we have now uploaded our code and data files to GitHub and this is reflected in the manuscript text.

I have now received two reviews of the manuscript “The optimal size of an ex situ conservation population: a comparison among 11 taxa in 5 genera.” Both reviewers thought that the manuscript would make a significant contribution and was of general interest; I agree.

Each of the reviewers has modest, but important, suggestions for changes that would strengthen the manuscript. I carefully read the manuscript and spent some time looking over the supplemental information. Based on my reading, I have identified additional areas where the manuscript could be strengthened. Based on the positive reviews, I request a revision that addresses each of the concerns raised.

Associate Editor Comments:

1) Where are the genetic results? Please provide a summary table with the results from the genetic analyses in the text or the supplemental material. The text reports second order results that come from the genetic analyses. The primary results need to be reported.

>>We now provide a summary table of standard genetic diversity analyses (number of samples per population, number of alleles, allelic richness, heterozygosity, FST) per wild population per species in the Supplementary Materials.

2) At slightly over 6 pages, about 50% of the manuscript, the Discussion is too long. Throughout, the length can be reduced, try for 1-2 pages. In particular, lines 385-407 and 411-460 seem like they could be cut a good bit and the conclusion is redundant with previous statements.

>>We agree the Discussion can be reduced. We have reduced the overall word count by more than 700 words, taking the Discussion from nearly seven pages to about five, including a large table (even including the suggested additions from the reviewers noted below). To do so we focused on deletions in the sections the Editor suggests, condensed our caveats without removing any essential points, rearranged a few sentences in the Discussion in order to reduce redundancy, and shortened the Conclusion paragraph. As some examples: several paragraphs reduced to a few sentences in lines 301-313, 340-345, and 409-421. We also deleted small phrases or single words throughout in order to shorten by a few more lines. We are confident the Discussion is now more concise and easy to extract the main points.

3) Table 3.

a) Elsewhere in the manuscript reduced and full data set results are reported. Please do that here as well.

b) The first column, “x times smaller” is awkward – people don’t usually use this phrasing. Recommend instead reporting the proportion of the current size that would meet the same result – this will resonate more easily with a reader.

>>a) **We have now included results for both Full and Reduced Datasets in Table 3 in the main manuscript (we had previously included both in the Supplemental but agree its best in the main document- thus we remove it from the Supplemental).**

>>b) **We now change this calculation to the editor's suggestion "proportion of current collection size" and explain this in the Table caption.**

4) line 130-131. Please provide a general summary of how the sampling and acquiring the genetic material were conducted in the text so the reader does not need to go to the supplemental section to understand how the research was conducted.

>>**We now explain briefly how tissue was collected, and laboratory and data procedures (Lines 136-140).**

5) line 145ff. This summary might have been more helpful at the beginning of the methods. The first 3 numbers in the list occur prior to the "computations" section that the list is placed in. Or else, perhaps it isn't really needed.

>>**We agree with the latter statement- it isn't really needed and we delete these items.**

6) line 171. Citation?

>>**We now provide a citation (CBD, 2012)- Convention on Biological Diversity, 2012. Global Strategy for Plant Conservation: 2011-2020. Botanic Gardens Conservation International, Richmond, UK**

7) line 178. Common alleles are >0.05 and low frequency alleles are between $0.01 < \text{low-freq} < 0.1$. So, between 0.05 and 0.1 alleles are both common and low frequency?? This seems awkward, suggest rethinking categories.

>>**Hoban and colleagues previously proposed the "low frequency" allele category as a bridge between common and rare alleles- these are rare enough that they are hard to catch, but not so rare as to be only a few copies in the population, while common enough to represent possibly alleles under diversifying selection or adaptations to periodic pressure such as disease (Hoban 2019, Hoban et al 2018, Hoban and Schlarbaum 2014). In order to retain comparability to recent work on this topic, we prefer that these categories should remain. Nonetheless we agree this could be explained better and we now put a sentence explaining the utility of this category in the Methods (Lines 191-193).**

8) line 277. What are GSPC and CBD?

>>**The abbreviations have now been spelled out**

9) Supplemental Material: I looked through the supplemental material a number of times. It is very difficult to assess what was done. I recommend reorganizing it to put all sampling together, all methods together, etc. This would also yield more consistency in the reporting for each of the taxa.

>>**We have now reorganized the Supplemental as advised- we put all the sampling text together (still separated by species) and all the genotyping methods together (still separated by species), and for the sampling we make sure to present tables in the same order for each species. Please note that because of such large rearrangement of text, Track Changes was not used on the Supplemental.**

10) Supplemental Material: provide exact locations of sampled wild material.

>>**These are all highly threatened species, and some are threatened by poaching or other exploitation. We therefore cannot provide locations for the palms and cycads. For the oaks, magnolias and hibiscus we can only provide locations to one decimal point due to concerns as well- this is now all in the Supplemental. We also provide the generic locations of the species native range in a Supplemental Figure.**

11) Supplemental Material: Read over the main Supplemental Material file with keen editorial eye. For example, remove word comments, at one point I noticed the text said "I" and not "we", try for consistency across the presentation of the different species.

>>**Thank you for noticing; this has been fixed and we read through it carefully**

Reviewer(s)' Comments to Author:

Referee: 1

Comments to the Author(s)

This well written article furthers the science related to producing generalized guidance for sample sizes of taxa for ex situ conservation, working across multiple genera. It is elegantly designed, particularly with reference to standard literature (ie. 95% of genetic diversity – Marshall and Brown) as well as current global targets (ie. 70% from the GSPC), and the binning/categories of common to rare alleles. Its main finding is that generalization by genus is not possible, although more general recommendations can be made. I provide a few comments below to improve the draft. I do think the article – including possibly the title – should be revised to qualify it further with regard to its specific focus (on recalcitrant, threatened woody perennials).

Lines:

16-21- probably need references for these international agreements

>>We have inserted web links to the full list of commitments of each of these agreements (Lines 17, 19 and 21)

23- Khoury et al. 2019 assess 3% for ex situ; 40% in situ; and almost 3% in combination. You might also note that this research did not measure genetic diversity directly but rather used geographic methods as proxy

>>We now include both the 3% and 40% statistics, and we explain the use of geography as proxy for genetic diversity (Lines 24-25)

26- although this article focuses on wild plants, I think it is worth noting here the very significant contribution of seedbanks (or genebanks) to genetic diversity conservation for species relevant to agrobiodiversity, particularly domesticated species.

>>We agree this is worth noting- we now mention "and with many seedbanks conserving agrobiodiversity, especially domesticated species." (Line 28)

32 I think that 'cannot' is too strong a word. Research increasingly shows, indeed, that most recalcitrant species can in fact be conserved given adequate research. The issue is more one of resource mobilization and priorities. Please consider an alternative like "are very challenging" or something like that. Note at 36 that its likely that cryo is an option for almost all species; again just a matter of resources and research.

>>We agree that this can be phrased better- we now change this to "are very difficult to conserve ex situ" and we note the resources/ expense involved in cryopreservation (Line 35 and 40)

77 and 96-102 - given the broad claim here, it will be extremely important to note the caveats and constraints that may not make the findings generalizable. Among potential others, these include: focus on perennial trees and shrubs (rather than across life forms in plants), focus on recalcitrant, the small (species) sample size of the study, etc. An alternative is to walk back the broad claims in the introduction and make it clearer what you are focusing on in the article (including in the title), e.g. on threatened woody perennials.

>>We agree the results need some constraint when generalizing them to all plants. We now mention the characteristics of our target species more clearly- their woody nature, long lives, and threatened status (Line 82, 278, and 498, also already mentioned in line 100). We also revisit this in the Discussion, specifically stating that our results should only be considered as representing such species (which would still be thousands of species, e.g. there are about 60,000 trees and at least 20% are threatened) in Line 478. We also specifically highlight the limitation of having only 11 species here for comparison in Lines 481-484.

162 why 75k?

>>We simply needed a large number of replicates. Actually a few thousand (or even few hundred) would likely have been sufficient as in previous work (Kashimshetty et al 2017, Hoban et al 2014, Hoban et al 2018, McGlaughlin et al 2015)

- **Kashimshetty, Y., et al., 2017. Effective seed harvesting strategies for the ex situ genetic diversity conservation of rare tropical tree populations. *Biodiversity and Conservation***
- **Hoban, S., et al., 2014. Comparative evaluation of potential indicators and temporal sampling protocols for monitoring genetic erosion. *Evolutionary applications***
- **Hoban, S., et al., 2018. Implementing a new approach to effective conservation of genetic diversity, with ash (*Fraxinus excelsior*) in the UK as a case study. *Biological conservation***
- **McGlaughlin, et al, 2015. How much is enough? Minimum sampling intensity required to capture extant genetic diversity in ex situ seed collections: examples from the endangered plant *Sibara filifolia* (Brassicaceae). *Conservation Genetics*,**

191 collector or collection?

>>**This could be “collector” as we are talking about the person doing the collecting or creating the conservation collection. However to make it more clear we will use “conservationist”.**

220 no need to capitalize Ex

>>**Agreed thanks for noticing this- fixed.**

Table 2- just a few questions for clarity- did you genotype every ex situ and every in situ plant noted in the table? And how certain are you that you captured all in situ plants/populations?

>>**We did genotype all individuals in this table, so we now include a note in the Table caption explaining this (Line 240). However as noted in the discussion, while we made strong efforts to find locations of the species we studied (Lines 131-133), we are not certain that we captured all in situ plants/ populations as there may have been populations we don't even know about. If some populations were missed, this would impact our estimates of capture of rare alleles of the species (explained in Lines 486-487)**

276- yes! But with very substantial resources and effort! Should be noted.

>>**We agree- We now mention the substantial resources involved (Lines 289-290)**

292 you might consider making this the title of the article, i.e. something like ‘Taxonomic similarity does not predict necessary sample size for genetic conservation (for threatened perennials- see earlier note)’

>>**Ok thank you for the suggestion- we now go with the following for a title: “Taxonomic similarity does not predict necessary sample size for ex situ conservation: a comparison among 5 genera”**

332 would it be possible to compute a third metric- the amount that collection size could go up or down in terms of sample size, to achieve ideal targets (70% or 95% or whatever)? This seems even more pertinent than the two metrics you currently provide.

>>**This might be possible. However to compute this would involve creation of a new methodology as there is no current approach for doing this in conservation genetics. Creating such a methodology is outside the scope of this paper but this seems like a good topic for a follow up study.**

356 yes and there are advantages to having more than one copy (i.e. a backup). It might be nice to note this more clearly and even recommend what a ‘safe’ genetic conservation strategy might look like (i.e. maybe 44 samples could conserve *Z. lucayana*, but perhaps each should be at least duplicated once (i.e. in two botanic gardens or something like that). At lines 429-437 same point; you might note that duplication could happen rather than sampling more plants.

>>**We fully agree this is an important point and we now provide more detail (lines 397-402)**

448- here is where you might mention that woody perennials (and recalcitrant ones at that) might not reflect overall plant diversity needs, and also the relatively small sample size of the study in terms of species/genera (even if its much larger than anything done before).

>>**We now include this note on the types of species we studied and that this does not apply to all plants. In the same paragraph we note the number of species and that more species should be studied to further explore this (Lines 478 and 482-484).**

Referee: 2

Comments to the Author(s)

This work represents a substantial contribution to our thinking about diversity in ex situ plant collections. The sampling and analyses are appropriate for your question and your results are clearly presented. I have provided some line specific comments below and I would recommend some minor additions/expansions in two areas:

1) Readers would benefit from more recommendations on what next steps should be based on your data. Would you recommend replacing/removing some collections from gardens that are redundant? Should there be more guidelines for how collections are made? Do you recommend more genetic work to better understand the drivers of diversity? Your results open up a lot of additional questions only some of which are addressed in the discussion.

>>Thank you for the suggestion. We have now included some recommendations on next steps, while keeping in mind the length of the Discussion. We add a couple sentences to the conclusion (now renamed Recommendations and Conclusions) about removal of some individuals in gardens based on a few criteria. We provide a simple summary of guidelines, and we do recommend more genetic work (Lines 482-484, 511, 513-519). Note that we also highlighted other next steps earlier in the discussion including similar work on more common species, on adaptive genetic variation, and on managing garden populations and metacollections for the long term (480-481, 490-493).

2) Making sure you are not overstating your results, specifically as it relates to organismal characteristics, range, collection effort. The conclusions you can draw related to necessary ex situ collection sizes to reach specific levels of diversity are very robust, but you lack insight into the drivers of wild diversity.

>>This is a good point and similar to one made by reviewer 1 about being more clear about what conclusions we can draw. We therefore add the caveats that our results might not apply to all species especially non-woody, non-perennials, and we add a sentence reminding that we have only looked at 11 species and therefore numerous other species should be examined (Lines 477-478, 482-484 for noting these issues, and also see below for response to reviewers final comment- we delete the statement about traits/ characteristics). We also add “sample size to sufficiently capture” to make it clear we are not referring to drivers of wild diversity (Line 336).

Specific Comments

Ln 87 – It is unclear what is meant by ‘threshold’. In later sections you make clear what thresholds you are examining, but it is unclear here.

>>We now clearly define that we are referring to the minimum number of plants to capture genetic diversity at two thresholds 70% and 95% (line 91)

Ln 104-124 – Please add family after each taxon.

>>Ok, done

Ln 157 – ‘possible sizes’ needs an additional description like ‘collection’

>>Agreed, and we now include this

Ln 272 – I count 6 species exceeding 70%

>>Yes, sorry for this oversight. It is correct, six species exceed 70% for the low frequency allele category. However, we are deleting this sentence in order to reduce length.

Ln 304-305 – It would be useful to provide more details related to demographic history. Did you explore potential bottlenecks? You have not provided any specific information about genetic diversity, so it is difficult to know the role it plays.

>>We did not test for bottlenecks in these species; here we only focus on the shape of the allele frequency distribution as an explanatory factor. Exploring bottlenecks and other aspects is a topic for further work.

However, in response to the Editors request, we do now provide specific information about genetic diversity and structure in the wild populations sampled in the Supplemental

Ln 306 – ‘minimum collection size’

>>Agreed. Thank you for the suggestion. However, we are deleting this sentence in order to reduce length

Ln 442 – This is an area where making more concrete recommendations would be useful.

>>As noted above, we now provide several recommendations here (Lines 482-484, 511, 513-519). Note that we also highlighted other next steps earlier in the discussion including similar work on more common species, on adaptive genetic variation, and on managing garden populations and metacollections for the long term (480-481, 490-493).

Ln 454 – Slightly more details on why you feel confident that most populations are represented in your dataset would be useful.

>>Agreed- we add notes on how we searched for populations to make it clear that most populations of each species are represented (already noted at Lines 131-133, explained in more detail Lines 486-487)

Ln 468-470 – This is somewhat overstated. I don't disagree with the listed traits being important, but you did not explicitly test any of them (F_{st} gets at some indirectly). You hypothesize that the identified traits/characteristics are important, but beyond genus not being a predictor you do not specifically test any of these.

>>We agree that this is overstated as we did not directly test these traits. We therefore delete this sentence and leave only the main conclusion in the previous statement regarding genus